# Characterization of the *Bax Inhibitor-1* Family in Cauliflower and Functional Analysis of *BobBIL4*

**DOI:** 10.3390/ijms25179562

**Published:** 2024-09-03

**Authors:** Xin Liu, Ning Guo, Shasha Li, Mengmeng Duan, Guixiang Wang, Mei Zong, Shuo Han, Zihan Wu, Fan Liu, Junjie Zhang

**Affiliations:** 1College of Life Science, Sichuan Agricultural University, No. 46, Xinkang Road, Ya’an 625014, China; lx18435990202@163.com (X.L.); 15733176623@163.com (S.L.); 2State Key Laboratory of Vegetable Biobreeding, National Engineering Research Center for Vegetables, Beijing Key Laboratory of Vegetable Germplasms Improvement, Key Laboratory of Biology and Genetics Improvement of Horticultural Crops (North China), Beijing Vegetable Research Center, Beijing Academy of Agriculture and Forestry Science, Beijing 100097, China; guoning@nercv.org (N.G.); duanmengmeng@nercv.org (M.D.); wangguixiang@nercv.org (G.W.); zongmei@nercv.org (M.Z.); hanshuo@nercv.org (S.H.); wuzihan20011224@126.com (Z.W.)

**Keywords:** cauliflower, Bax inhibitor-1, expression pattern, *BobBIL4*, abiotic stress

## Abstract

The *Bax inhibitor-1* (*BI-1*) gene family, which is important for plant growth, development, and stress tolerance, remains largely unexplored in cauliflower. In this study, we identified and characterized cauliflower *BI-1* family genes. Based on aligned homologous sequences and collinearity with Arabidopsis genes, we identified nine cauliflower BI-1 genes, which encode proteins that varied in length, molecular weight, isoelectric point, and predicted subcellular localization, including the Golgi apparatus, plasma membrane, and various compartments within the chloroplast. Phylogenetic analyses detected evolutionary conservation and divergence among these genes. Ten structural motifs were identified, with Motif 5 found to be crucial for inhibiting apoptosis. According to the cis-regulatory elements in their promoters, these genes likely influence hormone signaling and stress responses. Expression profiles among tissues highlighted the functional diversity of these genes, with particularly high expression levels observed in the silique and root. Focusing on *BobBIL4*, we investigated its role in brassinosteroid (BR)-mediated root development and salt stress tolerance. *BobBIL4* expression levels increased in response to BR and salt treatments. The functional characterization of this gene in Arabidopsis revealed that it enhances root growth and salinity tolerance. These findings provide insights into *BI-1* gene functions in cauliflower while also highlighting the potential utility of *BobBIL4* for improving crop stress resistance.

## 1. Introduction

Cauliflower (*Brassica oleracea var. botrytis*), which belongs to the family Brassicaceae, is characterized by an enlarged inflorescence (i.e., curd) that serves as its primary edible organ [1]. Notably, cauliflower is cultivated globally and widely consumed as a vegetable because of its exceptional nutritional value. Its curd comprises short, thick, and fleshy stems along with an undifferentiated inflorescence meristem, making it a distinctive feature of the plant. Cauliflower curd growth and development exhibit unique reproductive characteristics that are extremely sensitive to environmental changes. This sensitivity can lead to premature flowering, the formation of loose or “hairy” curds, and other developmental defects that adversely affect curd quality, potentially resulting in significant yield losses or even total crop failure. Consequently, compared with the cultivation of other commercially produced vegetables, the cultivation of cauliflower faces greater risks and challenges. Therefore, the genes and associated epigenetic and physiological mechanisms regulating cauliflower responses to abiotic stresses should be identified and characterized. Elucidating these underlying mechanisms is crucial for understanding how cauliflower adapts to environmental changes. This knowledge has significant implications for breeding new varieties with enhanced stress resistance, which is essential for stabilizing yield and quality under variable environmental conditions. Developing stress-resistant varieties will mitigate production risks while also ensuring the sustainable cultivation of cauliflower, ultimately benefiting consumer health.

Programmed cell death (PCD) is a widespread phenomenon in plants and animals [2]. While PCD is primarily known for its role in responses to various biotic and abiotic stresses, it also plays a general role in certain aspects of plant growth and development [3]. Apoptosis, a specific form of PCD, refers to a gene-regulated systematic process involving cellular self-destruction and is characterized by the loss of euchromatin structural information, leading to chromatin condensation and cell death [4]. More specifically, PCD in plants generally necessitates the activation of certain genes [5]. The Bax inhibitor-1 (*BI-1*) family of highly conserved transmembrane proteins has been extensively studied regarding its contributions to cell protection, ion homeostasis regulation, and anti-apoptotic activities [6]. *BI-1* family genes encode critical inhibitors of apoptosis that are responsive to endogenous and exogenous stimuli [7]. These genes have been thoroughly studied in terms of their effects on stress tolerance in various species. They are crucial for enhancing the tolerance of different organisms to a wide range of environmental stresses. Earlier research indicated the heterologous expression of Arabidopsis *BI-1* increases the tolerance of monocotyledonous and polycarpic plants to menadione sodium bisulfite, hydrogen peroxide, and drought-induced stress [8,9,10]. Under water-stress conditions, the expression of *AtBI1* regulates PCD in root tips [11]. In barley, *BI-1* is reportedly responsive to an infection by *Blumeria graminis*, resulting in compromised osmotic homeostasis [12]. Additionally, melatonin increases *BI-1* expression levels in *Medicago sativa* roots under highly saline conditions [13]. Arabidopsis contains five *BI-1* homologs, which have been designated as *LIFEGUARD 1-5* (*LFG1-5*). Both *LFG1* and *LFG3* inhibit apoptosis induced by endoplasmic reticulum (ER) stress, after which they participate in the inositol-requiring enzyme 1 (IRE1) signaling pathway to enable the plant to resume normal growth [14]. These two proteins also interact with membrane-bound progesterone receptor 3 (MAPR3) to modulate ER stress-induced apoptosis mediated by the IRE1 pathway [14,15,16]. In contrast, *LFG5* inhibits apoptosis through the IRE1 signaling pathway by regulating the balance of oxidized and reduced glutathione (GSH) [17]. Hence, *BI-1* family genes are important for plant resistance to abiotic and biotic stresses. However, *BI-1* gene families in horticultural crops have not been comprehensively investigated. Specifically, there has been relatively little research on the cauliflower *BI-1* gene family and its role in abiotic stress resistance. Among the *BI-1* family genes, *LFG2*, which is also known as *BRZ-INSENSITIVE-LONG HYPOCOTYLS 4* (*BIL4*), encodes a protein with seven transmembrane domains. The considerable interest in this gene is due to its pivotal role in plant processes, especially responses to brassinosteroid (BR) and various abiotic stresses. *BIL4* positively regulates BR signaling by preventing the degradation of the BR receptor BRI1, thereby ensuring plants are appropriately responsive to BR, which is necessary for optimal growth and adaptations to environmental changes [18]. A previous study showed that *BIL4* is essential for maintaining cell turgor and overall stress tolerance, suggesting that it helps to preserve cellular structures critical for stress tolerance [18]. *BIL4* may interact with other stress-responsive genes to enhance the ability of plants to cope with adverse environmental conditions [19]. In our previous comparative genomic study, we identified structural variations in the *BIL4* promoter region between cauliflower and cabbage, implying that *BIL4* may differentially affect various *B. oleracea* morphotypes [20]. This raises several important questions: What are the characteristics and tissue-specific features of the *BI-1* gene family in cauliflower? How do these genes (e.g., *BIL4*) modulate cauliflower growth and development, particularly in response to abiotic stress?

In this study, nine *BI-1* homologs were identified in cauliflower via BLAST and collinearity analyses. A bioinformatics analysis characterized these genes in terms of their properties, phylogenetic relationships, chromosomal locations, structures, conserved motifs, and predicted cis-acting elements. In addition, *BobBIL4* was expressed specifically in the ER membrane in the roots, cotyledons, petals, and apical regions of siliques. Furthermore, *BobBIL4* expression was responsive to BR and resulted in substantial abiotic stress tolerance. These findings offer valuable scientific and practical insights relevant to future research on the biological role of *BobBIL4* in cauliflower, with implications for clarifying the evolution of stress tolerance in *Brassica* species.

## 2. Results

### 2.1. Identification of Cauliflower BI-1 Family Genes

Using the Arabidopsis *BI-1* gene family for a sequence homology analysis, we identified nine *BI-1* family genes in cauliflower (Appendix A). These genes were named according to the corresponding Arabidopsis genes. Specifically, the *AtLFG1-4* Arabidopsis genes had one homologous gene in cauliflower, whereas *AtLFG5* had three homologous genes, which resulted from whole-genome duplication and tandem duplication events. Additionally, two homologous genes of *AtBI-1* were identified in cauliflower. The characteristics of the proteins encoded by the nine cauliflower *BI-1* family genes were analyzed, revealing lengths ranging from 239 to 256 amino acids, molecular weights between 26.98 and 30.94 kDa, and isoelectric points (pI) between 5.88 and 9.05. The instability index varied from 24.72 to 46.95, the hydrophilicity scores ranged from 0.474 to 0.932, and the lipid index was between 107.77 and 130.50. These proteins were predicted to be predominantly localized in the Golgi apparatus, plasma membrane, cell membrane, and chloroplasts. Notably, these BI-1 family proteins were revealed to contain seven transmembrane domains and no signal peptides (Table 1). In terms of their chromosomal distribution, the nine cauliflower BI-1 family genes were detected on chromosomes C01, C02, C04, C05, C07, C08, and C09, of which chromosomes C02 and C07 each contained two genes, whereas the remaining five chromosomes contained one gene each (Appendix A).

### 2.2. Phylogenetic Analysis of BI-1 Family Genes

To investigate the evolutionary relationships of *BI-1* family genes, a phylogenetic tree was constructed using the amino acid sequences encoded by the nine *BI-1* genes identified in cauliflower as well as the amino acid sequences of six, seven, and four *BI-1* proteins in Arabidopsis, rice, and maize, respectively [17,21] (Figure 1). In the phylogenetic tree, the cauliflower *BI-1* gene family was divided into five groups. Most cauliflower *BI-1* family genes were clustered with their Arabidopsis homologs in the same group, suggesting evolutionary conservation within this gene family. Intriguingly, *BobLFG1* in cauliflower was not clustered with *AtLFG1* in Arabidopsis, indicative of a possible sequence variation and functional divergence. Sequence-based phylogenetic analyses offer valuable insights for further functional analyses of the *BI-1* gene family in cauliflower.

### 2.3. Structural and Conserved Motif Analyses of Cauliflower BI-1 Genes

To assess the diversity in protein structures, we used Hidden Markov Models, CDD, and MEME to analyze the conserved motifs and domains of BI-1 proteins. Ten motifs were predicted within BobBI-1 amino acid sequences. Motif 5 (EKDKKKEKK) was shared by all nine BobBI-1 proteins (Figure 2A,B,D). This motif, which is located at the C-terminal end of most proteins, is crucial for inhibiting apoptosis induced by extensive stimulation [22,23]. BI-1 proteins contain seven transmembrane domains [24]. Additionally, the exon–intron organization of the nine cauliflower *BI-1* genes was analyzed to gain insights into gene structures (Figure 2C). Both *BobLFG5.1* and *BobLFG5.2* comprised three exons, whereas *BobBIL4*, *BobLFG4*, *BobLFG5.3*, and *BobLFG3* contained four exons. In contrast, six exons were detected in each of the remaining *BI-1* genes. The diversity in the number and distribution of exons and introns among the *BI-1* genes reflects a complex and heterogeneous genomic landscape.

To functionally characterize *BobBI-1* genes and gain insights into the precise regulation of their stress-responsive expression, we analyzed the promoter cis-elements within the 2 kb sequence upstream of the ATG start codon of *BobBI-1* genes. All *BobBI-1* promoters contained elements responsive to various hormones (e.g., methyl jasmonate, gibberellin, salicylic acid, abscisic acid, and auxin) as well as elements responsive to light. Specifically, the *BobLFG1* promoter included regulatory elements related to anaerobic induction and low-temperature responsiveness. The *BobLFG3* promoter contained elements associated with anaerobic induction and expression in differentiated tissues. Furthermore, elements related to anaerobic induction, stress response, cold response, and hypoxia-specific induction were detected in the *BobBI1.2* promoter. The promoter of *BobBIL4* contained elements related to anaerobic, stress, and hypoxia-specific induction as well as circadian rhythm control. The *BobLFG4* and *BobLFG5.1* promoters included drought response-related elements. The *BobBI1.1* promoter harbored elements associated with stress responses, including those specific to anaerobic conditions. Similarly, the *BobLFG5.2* promoter contained elements related to anaerobic induction, stress induction, and differentiated tissue expression. The promoter of *BobBI1.2* contained elements associated with anaerobic induction and low-temperature responsiveness. Only *BobLFG5.3* and *BobLFG5.2* had promoters with MYB transcription factor-binding sites and were involved in anthocyanin synthesis (Appendix A). In addition, the promoters of all family members, except for *BobLFG3*, contained G-boxes. With the exception of *BobBI1.1*, all family members had promoters comprising Box-4 elements. The *BobBI-1* promoters were particularly rich in stress-responsive elements (42 in total). In contrast, there were relatively few light-responsive (4CK-CMA1b) elements and elements related to tissue-specific expression (RY-elements) (i.e., one each). These findings suggest that cauliflower *BobBI-1* genes may encode key regulators of hormone signaling pathways, stress responses, and growth and development.

### 2.4. BobBI-1 Expression Profiles in Different Cauliflower Tissues

To investigate the potential differences in cauliflower *BI-1* gene functions, we analyzed the expression patterns of the nine *BI-1* family genes in various tissues (e.g., root, stem, leaf, curd, bud, flower, and silique) using transcriptome data [20]. *BobLFG3* was expressed at low levels in all examined tissues, with no detectable expression in the root, stem, leaf, or flower. This suggests that in cauliflower, *BobLFG3* may play a very limited role under normal growth conditions. *BobLFG1* was highly expressed only in the silique. *BobLFG5.3* had the highest average expression level in all tissues, with the highest expression levels in each tissue. Additionally, its two homologous genes (*BobLFG5.1* and *BobLFG5.2*) were also highly expressed in all tissues. Hence, these three homologous *LFG5* genes, which were retained after a whole-genome triplication event, likely have important functions in cauliflower. The *BobLFG4* expression level, which was relatively low in most tissues, was highest in the flower. The average *BI-1* expression level was highest in the silique, followed by the root, and was lowest in the leaf. The variability in the expression of these genes was greatest in the root. *BobLFG5.1*, *BobLFG5.2*, and *BobLFG1* were expressed at relatively high levels in the silique, whereas *BobBIL4*, *BobLFG5.3*, *BobBI1.1*, and *BobBI1.2* were most highly expressed in the root (Figure 3A–C). Correlation analyses of gene expression revealed the following significant correlations among the nine *BI-1* genes: *BobBI1.2* with *BobBIL4*, *BobLFG5.1*, and *BobLFG5.2*; *BobLFG1* with *BobLFG3*; and *BobBI1.1* with *BobLFG5.3* (Figure 3D). These findings suggest that the nine *BI-1* family genes are differentially expressed and have undergone functional differentiation in cauliflower. Moreover, the gene expression data provide a theoretical basis for future studies on gene functions.

### 2.5. Analyses of BobBIL4 Expression and Subcellular Localization of the Encoded Protein

In plants, *BIL4* is critical for responses to BR and stress resistance. Additionally, structural differences in the *BIL4* promoter region revealed by a comparison between cauliflower and cabbage may be related to cauliflower development and stress resistance. To further functionally characterize *BobBIL4* in cauliflower, we analyzed its expression pattern in various Arabidopsis tissues using a *ProBobBIL4-GUS* reporter line. We also investigated the subcellular localization of BobBIL4 in tobacco leaf epidermal cells. According to the detection of GUS signals, *ProBobBIL4* was active in transgenic Arabidopsis root tips, cotyledons, leaves, inflorescences, and siliques (Figure 4A–E). In tobacco leaf epidermal cells, the green fluorescence due to the transient expression of an eGFP-BobBIL4 fusion protein overlapped with the red fluorescence from the RFP-HDEL fusion construct (i.e., ER marker) (Figure 4F,G). This observation is consistent with the findings of an earlier study that determined that BI-1 is predominantly localized in the ER [25].

### 2.6. BobBIL4 Plays a Pivotal Role in BR-Mediated Root Development

Earlier research revealed the crucial regulatory effects of BR on plant root development (e.g., promoting root growth, root hair formation, and lateral root initiation), enabling roots to adapt to environmental stimuli and enhancing stress tolerance [26,27,28,29]. Previous studies in Arabidopsis have indicated that brassinosteroids (BRs) exhibit spatiotemporal dynamics in root tissues. In the current study, *BobBIL4*, *BobLFG5.3*, *BobBI1.1*, and *BobBI1.2* were highly expressed specifically in the cauliflower root. To determine if these genes are responsive to BR in the root, we analyzed the changes in their expression within 12 h after a 24-epibrassinolide (EBL) treatment. The elongation of Arabidopsis roots is highly sensitive to BR concentration, with EBL being one of the most bioactive brassinosteroid biosynthetic intermediates. There was a continuous increase in the *BobBIL4* expression level following the EBL treatment, with a significant increase detected after 8 h and peak expression at 12 h. The expression of *BobLFG5.3* was affected soon after the EBL treatment, with a significant increase at 2 h, peak expression at 8 h, and a decrease at 12 h. The *BobBI1.1* expression level was highest at 2 h post-treatment, after which it decreased. In contrast, *BobBI1.2* expression decreased significantly after the EBL treatment. These results suggest that although these four genes were highly expressed in the root, they are differentially responsive to BR signals. Notably, *BobBIL4* expression in the root increased continuously in response to BR (Figure 5A).

To further investigate the effects of *BobBIL4* on BR-mediated root development, we utilized wild-type Arabidopsis (Columbia) plants, *bil4* mutants (in which the endogenous *BIL4* gene is knocked out), and Arabidopsis lines complemented with *BobBIL4* under the control of the 35S promoter in the *bil4* mutant background. As detailed in the Section 4, the expression levels of *BIL4* in the complemented lines *BobBIL4* in *bil4* #2 and *BobBIL4* in *bil4* #7 are approximately 90 times higher than in the wild type. Consequently, we have designated these transgenic lines as *BobBIL4*-overexpressing lines/plants. The inhibitory effects of the BR antagonist Brassinazole (Brz) on BR biosynthesis were examined. Brz, a known suppressor of BR synthesis, targets plant hormones crucial for proper developmental processes. Seeds were sown on solid MS medium containing 2 µM EBL or 3 µM Brz to examine root development in 3-day-old seedlings. On the normal MS medium, *bil4* mutant plants had significantly shorter roots than wild-type plants. The expression of cauliflower *BobBIL4* in *bil4*-deficient lines restored normal root development. The EBL and Brz treatments of the wild-type, *bil4* mutant, and *BobBIL4*-overexpressing plants inhibited root growth and development, although the extent of the inhibition varied. Following the EBL treatment, root growth was inhibited the most in the *bil4* mutant (according to the average root length). However, the average root length of the *BobBIL4*-overexpressing lines was similar to that of the wild-type control, indicating that *BobBIL4* expression may mitigate the inhibitory effects of exogenous BR on root growth (Figure 5B,C,E). The addition of Brz to the MS medium inhibited root development in the examined genotypes, but the differences were not significant, suggesting that *BobBIL4* likely regulates root growth and development through BR (Figure 5B,D,E). These findings highlight the importance of *BobBIL4* for BR-mediated root development.

### 2.7. BobBIL4 Expression Enhances Salt Stress Tolerance

To clarify the effects of salt stress on the expression of *BobBIL4*, *BobLFG5.3*, *BobBI1.1*, and *BobBI1.2*, which are relatively highly expressed in the root, we analyzed the changes in their expression within 12 h after a 100 mM NaCl treatment. The expression levels of *BobBIL4* and *BobBI1.2* increased significantly after the salt treatment. More specifically, their expression levels initially increased, peaking at 4 h post-treatment, and then decreased. Interestingly, the extent of the expression level change was greater for *BobBIL4* than for the other genes. In contrast, *BobBI1.1* expression was relatively unchanged under saline conditions, suggesting it may not play a crucial role in the salt stress response. The expression of *BobLFG5.3* increased significantly at 4 h post-treatment but decreased at 12 h (Figure 6A). These results indicate that these four genes are differentially responsive to salt stress, with *BobBIL4* exhibiting the most rapid and significant response.

To further analyze the contribution of *BobBIL4* to the root defense against salt stress, wild-type Arabidopsis plants, *bil4* mutants, and *BobBIL4*-overexpressing lines with a *bil4*-deficient genetic background were grown on MS medium containing 100 mM NaCl to compare the root development of 9-day-old seedlings. Under normal growth conditions, *bil4* mutant plants had significantly shorter roots than wild-type plants and *BobBIL4*-overexpressing lines, implying that *BIL4* plays an essential role in root growth. Moreover, *BobBIL4* expression is conducive to restoring normal root development in the Arabidopsis *bil4* mutant. The 100 mM NaCl treatment resulted in shortened roots in all four genotypes, indicative of inhibited root development. However, the inhibitory effects of the salt treatment on root growth were weakest for the *BobBIL4*-overexpressing lines (Figure 6B–D). Based on electrical conductivity, we assessed membrane permeability, which indicated that cellular damage was most severe in the *bil4* mutant, whereas the cellular damage of the *BobBIL4*-overexpressing lines was similar to that of the wild-type control (Appendix A). This finding suggests that *BobBIL4* expression can enhance the salt stress tolerance of Arabidopsis, highlighting its critical role in the plant response to salinity.

## 3. Discussion

During ontogenic and morphogenetic development, plants are exposed to multiple stressors; the detrimental effects of these stressors on intracellular homeostasis are mitigated by complex regulatory networks [30,31]. The *BI-1* gene family plays a crucial role in modulating cellular stress responses and maintaining homeostasis in animal and plant species exposed to biotic and abiotic stressors [32]. This gene family has been detected in many important crops, including *Triticum aestivum* [33], *Daucus carota* [34], and *Brassica napus* L. [35]. However, comprehensive investigations on the functions and phylogenetic relationships of this gene family have not been conducted. In this study, we identified and characterized nine *BI-1* genes in the cauliflower genome. Using a range of bioinformatics approaches, we analyzed these genes and their transcriptional profiles across various tissues and in response to abiotic stressors. The *BobBIL4* response to stress was validated via heterologous expression in Arabidopsis. The results of these analyses indicate that the *BobBI-1* sequence is relatively conserved, and its expression patterns vary slightly in different tissues and under various stress conditions. The study findings provide the foundation for future investigations on the biological functions of *BobBIL4*.

Adaptations to environmental changes may involve alterations to exon–intron structures in genes [36]. Analyzing gene structures can reveal phylogenetic relationships and elucidate the evolution of gene families [37]. In the present study, analyses of *BobBI-1* genes detected conserved domains (Figure 3C) as well as variations in exon–intron architecture and motifs (Figure 3B), suggestive of diverse biological roles. Moreover, although *BI-1* genes were revealed to contain conserved sequences, which implies functional similarities, they differed in terms of tissue-specific expression and stress-induced expression. The characterization of the cauliflower *BI-1* gene family has enhanced our understanding of stress tolerance-related mechanisms, with potential implications for optimizing crop tolerance to environmental stresses.

Identifying cis-acting elements is a standard approach to investigating temporospatial gene expression patterns as well as gene expression associated with tissue growth and development [38]. The cis-elements located within promoter regions are critical for controlling transcription and often serve as binding sites for specific transcriptional regulators [39,40]. Earlier research showed that *BI-1* genes mediate adaptations to abiotic and biotic stressors, especially physiological or environmental conditions that induce ER stress [41]. In Arabidopsis, *AtBI-1* is expressed in a tissue-specific manner, *AtLFG1* is highly expressed in young leaves, during the floral development period, and in tender siliques, and *AtLFG2* (*AtBIL4*) is highly expressed in tender tissues and developing flowers; this gene also influences hypocotyl cell elongation via the oleuropein lactone signaling pathway [18,21]. *AtLFG3* is highly expressed during seed germination and floral development stages, but its expression is also significantly upregulated under ER and salt stress conditions [15,16]. In the present study, *BobBI-1* expression differed slightly among cauliflower tissues. In addition, *BobBIL4*, *BobLFG5.3*, *BobBI1.1*, and *BobBI1.2* were highly expressed in the root. Moreover, the tissue-specific expression pattern of *BobBIL4* is consistent with that of *AtBIL4*. Our analysis of subcellular localization indicated that most of the examined *BI-1* genes encode proteins primarily located in the ER membrane and Golgi apparatus. We also confirmed the subcellular localization of BobBIL4 in the ER membrane (Figure 4E–J). Future studies on *BobBIL4* should focus on the effects of its expression in root tissue on membrane protein dynamics.

Apoptosis, as a form of programmed cell death, not only involves the systematic self-destruction of cells but is also marked by the progressive loss of euchromatin structural information. This loss contributes to chromatin condensation, a key feature of the apoptotic process, and underscores the complex epigenetic regulation underlying apoptosis.

The expression of *BI-1* genes can be induced by various stresses, indicating that these genes may differentially contribute to stress tolerance [23]. In *D. carota*, the overexpression of *HvBI-1* in the root confers resistance to nematodes [34]. The overexpression of *AtBI-1* enhances plant tolerance to various cell death-associated stresses. For example, the heterologous expression of *AtBI-1* increases the drought tolerance of transgenic sugarcane plants [10], whereas it enhances the adaptation of transgenic rice plants to oxidative stress [8]. Furthermore, PCD is affected by multiple plant hormones. A recent study showed that jasmonic acid modulates the expression of PCD-related genes in salt-stressed rice seedlings [42]. Both *TaBI1.1* and *AtBI-1*, which have highly conserved sequences, positively affect heat stress tolerance [43]. In the current study, the expression levels of four *BI-1* family genes increased in the root following BR and salt treatments. The distinct expression profiles of these genes under different stress conditions may reflect their diverse roles associated with plant adaptations to stress. In darkness, the *bil4* mutant is insensitive to Brz; this mutant was previously compared with *BIL4*-overexpressing seedlings in terms of hypocotyl elongation [21]. Consistent with the findings of an earlier study [44], we determined that BR inhibits root elongation under light. Additionally, after a Brz treatment, the *BobBIL4*-overexpressing lines had the shortest roots, which was in accordance with the findings of previous research on hypocotyl elongation [18,21]. The enhanced tolerance of *BobBIL4*-overexpressing lines to salt stress suggests *BobBIL4* may be useful for developing stress-resistant crops. However, it is important to acknowledge certain limitations in our study related to the use of the 35S promoter, which drives constitutive expression across all cell types and developmental stages. This approach, while effective for broad analysis, may not fully reflect the natural expression pattern of *BobBIL4*, which could be restricted to specific cell types or developmental stages. The resulting phenotypic effects observed in our study might differ from those under more natural expression conditions. Future research should focus on identifying and using cell type- or stage-specific promoters to better mimic *BobBIL4*’s natural expression context, providing a more accurate understanding of its physiological roles. Additionally, further studies are needed to explore the function of *BobBIL4* and its homologs in stress responses, particularly in cauliflower, to develop more targeted strategies for enhancing stress tolerance in crops.

## 4. Materials and Methods

### 4.1. Identification of BobBI-1 Gene Family Members

The “Korso” cauliflower reference genome (https://figshare.com/collections/Korso_and_OX_heart_genome_assemblies_and_annotations/5392466, accessed on 12 January 2024) was used to identify *BobBI-1* family genes in cauliflower [20]. PF01027 (Bax1) (http://pfam.xfam.org/, accessed on 12 January 2024) from the Pfam database was employed to identify *BI-1* gene family members. Amino acid sequences were obtained from Arabidopsis and cauliflower databases and analyzed using the NCBI Conserved Domain Database (CDD) (https://www.ncbi.nlm.nih.gov/, accessed on 12 January 2024). Homology models were constructed using SMART (v8.0) (http://smart.embl-heidelberg.de/, accessed on 12 January 2024.), while incomplete sequences were filtered using InterPro (v5.52-86.0) (https://www.ebi.ac.uk/interpro/, accessed on 12 January 2024) [45].

### 4.2. Determination of Biochemical Properties

The *BI-1* family genes were examined regarding their length as well as the amino acid composition, molecular weight, isoelectric point (pI), and hydrophilicity of the encoded proteins using ExPaSy ProtParam (https://web.expasy.org/protparam/, accessed on 24 February 2024) [46]. Additionally, protein subcellular localization was predicted using the Cell-PLoc 2.0 suite [47].

### 4.3. Chromosomal Localization of Cauliflower BI-1 Family Genes

For the chromosomal mapping of *BI-1* genes, we used TBtools to analyze the GFF3 data for the cauliflower genome. Subsequently, gene repetition was examined using MCScanX (v2.034), with the results visualized using TBtools (v2.034) [48].

### 4.4. Phylogenetic Analyses

Six, seven, and four *BI-1* sequences from Arabidopsis, rice, and maize, respectively, were retrieved from the Ensembl Plants database (https://plants.ensembl.org, accessed on 16 March 2024) and aligned with the cauliflower BI-1 sequences using MUSCLE within MEGA (v11) [49]. On the basis of the aligned sequences, a phylogenetic tree was constructed according to the neighbor-joining method with 1000 bootstrap replicates. The tree was visualized using EvolView (v2.0) (https://evolgenius.info//evolview-v2/#mytrees/1/2, accessed on 24 February 2024) [49].

### 4.5. Analyses of Gene Structures, Conserved Motifs, and Conserved Sequences

The exon–intron architecture of genes was determined using GSDS2.0 (http://gsds.gao-lab.org/, accessed on 24 May 2024), whereas conserved motifs were identified using the MEME platform (https://meme-suite.org/meme/tools/meme, accessed on 24 May 2024), configured to identify 10 motifs [50]. In addition, DNAMAN was used for the comparative analysis of BI-1 protein sequences. Results were visualized using TBtools (v2.034).

### 4.6. Analysis of Cis-Acting Elements

PlantCare was used to screen the 2 kb sequence upstream of *BI-1* genes for cis-acting elements [51], and these elements were visualized using TBtools (v2.034).

### 4.7. Gene Duplication and Covariance Analyses

According to the chromosomal locations of *BI-1* genes, intra- and inter-genic covariances among cauliflower and Arabidopsis *BI-1* genes were analyzed using MapChart (v2.0) [52]. MCScanX (v2.034) was used to examine segmental duplication events and homologous relationships.

### 4.8. Subcellular Localization of BobBIL4

The open reading frames (ORFs) of BobBIL4 were cloned into pFGC-eGFP for the subsequent transformation of tobacco (*Nicotiana benthamiana*) epidermal cells. Seedlings were grown in an insect-free chamber at 25 °C with an 8 h light:16 h dark cycle. Epidermal cells were transformed with the recombinant plasmid using Agrobacterium tumefaciens GV3101. For the co-expression of eGFP-BobBIL4, RFP-HDEL, and p19, bacterial cultures were mixed at a 1:1:1 ratio and incubated at room temperature for 1–3 h prior to the infiltration of tobacco at the 5- to 6-leaf stage using a 1 mL syringe. Fluorescence was captured at 48 h post-infiltration using a LSM780 (Carl Zeiss, Jena, Germany), with settings adjusted for detecting fluorophores.

### 4.9. Preparation of Plant Materials and Stress Treatments

Stress treatments were completed using cauliflower “Korso” plants [20]. Seeds were surface-sterilized and germinated on solid Murashige and Skoog (MS) medium for 6 days in a temperature-controlled incubator. Uniformly growing seedlings were selected for the following liquid MS medium-based stress treatments (0, 2, 4, 8, and 12 h): 2 μM 24-epibrassinolide (EBL) or 3 μM brassinazole (Brz) (hormones) and 200 μM NaCl (salinity). Root samples were harvested, flash-frozen in liquid nitrogen, and stored at −80 °C until analyzed.

Arabidopsis thaliana Columbia-0 ecotype seeds were preserved in the laboratory. The *bil4* mutant (Salk_052507C) was obtained from the Salk Institute T-DNA collection, with the T-DNA insertion verified by PCR using plant genomic DNA and primers flanking the T-DNA and gene-specific sequences. Transgenic Arabidopsis was generated via floral dip transformation. For the wild-type Columbia background, we introduced the *BobBIL4* promoter (*ProBobBIL4*) fused with the GUS reporter gene to analyze tissue-specific expression patterns. In the *bil4* mutant background, we introduced the coding sequence (CDS) of *BobBIL4* under the control of the 35S promoter to complement the *bil4* mutation and further investigate the functional role of *BobBIL4*. We utilized quantitative fluorescence PCR to measure the expression of the *BIL4* gene in wild-type plants and the *BobBIL4* gene in the complemented lines *BobBIL4* in *bil4* #2 and *BobBIL4* in *bil4* #7. The results showed that the expression levels of *BobBIL4* in the two transgenic lines were 84 and 90 times higher than the *BIL4* expression in the wild type, respectively. The *bil4* mutant line shows significantly reduced expression compared to the wild type, consistent with the findings of Wang et al. (2019) [16] (Appendix A). Seeds underwent stratification at 4 °C for a period of 2 days to induce germination. Subsequently, the plants were cultivated under continuous ½-strength Murashige and Skoog (MS) medium (22 °C, 16 h light/8 h dark cycle, and 60% humidity). To examine promoter activity patterns among tissues and developmental stages. Seedlings and the inflorescence organs of *proBobBIL4* were incubated for 12 h at 37 °C in the staining buffer, respectively.

For the salt stress experiments, 3-day-old seedlings with the same growth trend were transferred to a medium containing 100 mM NaCl, 2 μM EBL, and 3 μM Brz. After a 9-day treatment, root length was measured and photographed, and lon Leakage Measurement.

### 4.10. Ion Leakage Measurement

The permeability of the cell membrane was evaluated through the quantification of ion efflux from seedlings post-salinity treatment. The conductivity was assessed as previously reported [15]. Following each assay, 30 seedlings were submerged in 20 mL of distilled water with mild agitation for 2 h at room temperature. Each sample was subjected to triplicate biological replicates.

### 4.11. Total RNA Extraction and Expression Analysis

Total RNA was extracted using an RNAprep Pure Kit (Vazyme, Nanjing, China). RNA quality was assessed via 1% agarose gel electrophoresis, whereas the RNA concentration and purity were determined using a NanoDrop 2000 spectrophotometer (Thermo Fisher Science, Wilmington, NC, USA). First-strand cDNA was synthesized from 1 μg RNA using HiScript^®^ III All-in-One RT SuperMix (Vazyme) for the qRT-PCR analysis of gene expression using Taq Pro Universal SYBR qPCR Master Mix (Takara, Shiga, Japan) and a CFX96 real-time PCR instrument (BioRad, Shanghai, China). Gene-specific primers were designed using Integrated DNA Technologies online software (https://sg.idtdna.com/pages, accessed on 20 June 2024). *BobACTIN* (BolK_3g69850) was selected as an internal reference gene. The qRT-PCR analysis (three technical replicates per reaction) was completed using a reaction mixture comprising 10 μL Taq Pro Universal SYBR qPCR Master Mix, 0.5 μL forward/reverse primers, 1 μL cDNA template, and 8 μL ddH_2_O. The qRT-PCR program was as follows: 95 °C for 2 min, 39 cycles of 95 °C for 10 s, and 60 °C for 30 s. Relative expression levels were calculated according to the 2-∆∆Ct method [53].

Seven different tissues of Korso (root, stem, leaf, curd, bud, flower, and silique) were collected for transcriptome sequencing. The tissues were assessed as previously reported [20]. Tissue-specific expression patterns were visualized using heatmaps generated by R (v4.2) (https://www.r-project.org, accessed on 20 June 2024).

### 4.12. Statistical Analysis

The data analyzed were processed and visualized by GraphPad Prism 9.0 (GraphPad Software, San Diego, CA, USA) and shown in mean ± SD. Significant differences were identified using one-way ANOVA. A student’s t-test was used to calculate *p* values (* *p* < 0.05, *** *p* < 0.01, *** *p* < 0.001).

## 5. Conclusions

In this study, we identified and characterized the cauliflower *BI-1* gene family. The evolutionary conservation and divergence among these genes may reflect a complex interplay between functional redundancy and specialization that enables cauliflower to adapt to diverse external stresses. The critical roles of specific *BI-1* genes, such as *BobBIL4*, in hormone signaling and stress responses are relevant to future research aimed at improving crop resilience through genetic manipulation. Additional studies focusing on the mechanistic basis of *BI-1*-mediated stress responses are necessary to ensure these findings are exploited to improve the production of cauliflower and other agriculturally important crops.

## Figures and Tables

**Figure 1 ijms-25-09562-f001:**
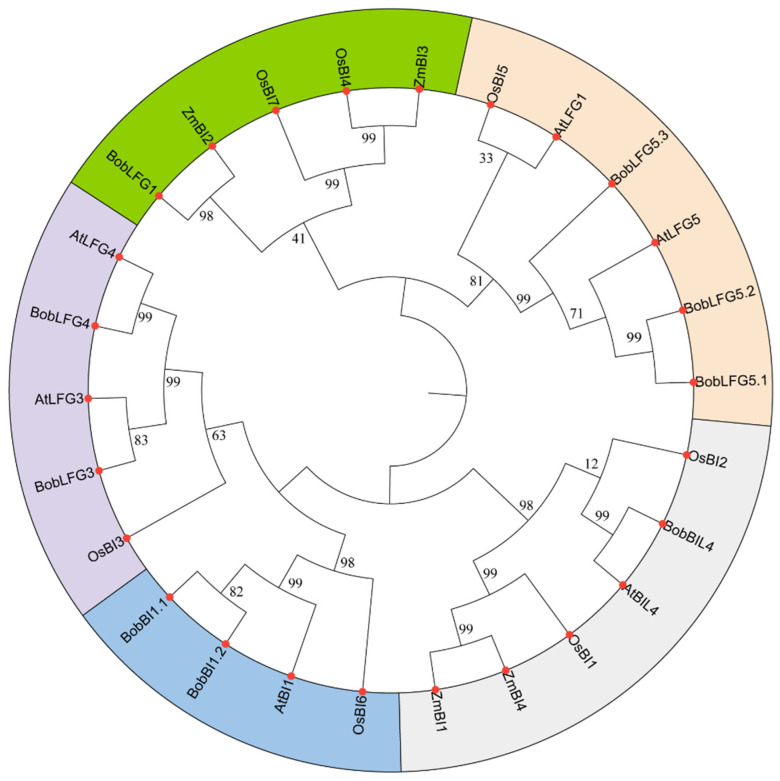
Phylogenetic tree of *BI-1* genes in cauliflower, Arabidopsis, rice, and maize. Neighbor-joining trees were generated using the MEGA (v11) software, with 1000 bootstrap replicates. All *BI-1* genes were classified into five subgroups, which are differentiated by color.

**Figure 2 ijms-25-09562-f002:**
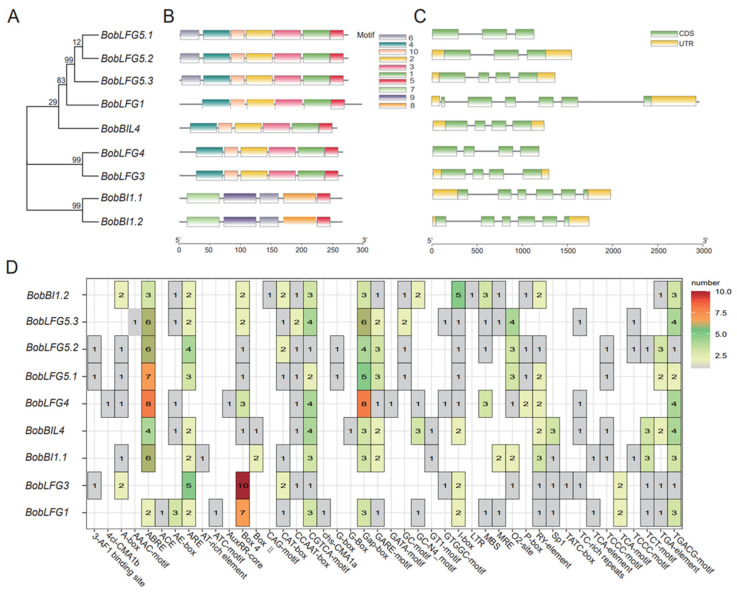
Structural characteristics of domains and promoter cis-regulatory elements of *BI-1* family genes in cauliflower. (**A**) Phylogenetic tree. (**B**) Conserved motifs. (**C**) Structure of *BobBI-1* genes. (**D**) Classification and number of promoter cis-acting elements, with the color intensity reflecting the number of cis-acting elements.

**Figure 3 ijms-25-09562-f003:**
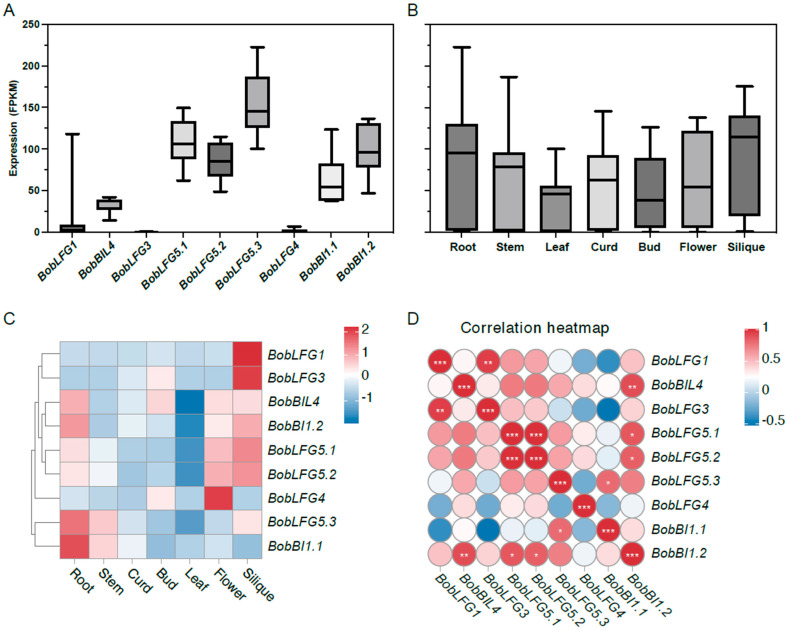
Expression profiles and correlation analysis of *BI-1* family genes in cauliflower across different tissues. (**A**) Boxplot showing the expression levels (FPKM) of nine *BI-1* family genes in cauliflower across various tissues, including root, stem, leaf, curd, bud, flower, and silique. (**B**) Boxplot depicting the overall expression distribution of the nine *BI-1* family genes in each tissue type, indicating differential gene expression patterns across the tissues. (**C**) Heatmap illustrating the expression profiles of the nine *BI-1* family genes in different cauliflower tissues. The color scale represents the expression level, with red indicating higher expression and blue indicating lower expression. (**D**) Correlation heatmap showing the Pearson correlation coefficients between the expression levels of the nine *BI-1* family genes in different cauliflower tissues. * *p* < 0.05, ** *p* < 0.01, *** *p* < 0.001.

**Figure 4 ijms-25-09562-f004:**
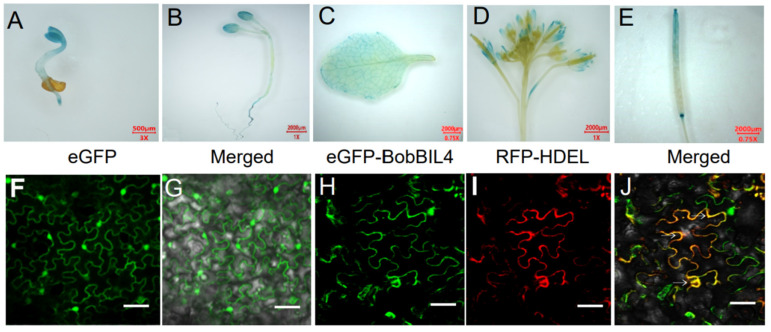
BobBIL4 expression patterns in Arabidopsis and subcellular localization. (**A**–**E**) Tissue-specific expression of BobBIL4 in transgenic Arabidopsis. (**F**–**J**) Subcellular localization of BobBIL4 in *Nicotiana benthamiana* leaf epidermal cells. Cells containing the empty vector pFGC-eGFP served as the control. (**F**) eGFP image. (**G**) Merged eGFP and bright field images. (**H**) Subcellular localization of eGFP-BobBIL4. (**I**) Endoplasmic reticulum marker RFP-HDEL. (**J**) Merged eGFP-BobBIL4, RFP-HDEL, and bright field images. Scale bars = 50 μm (**F**–**J**).

**Figure 5 ijms-25-09562-f005:**
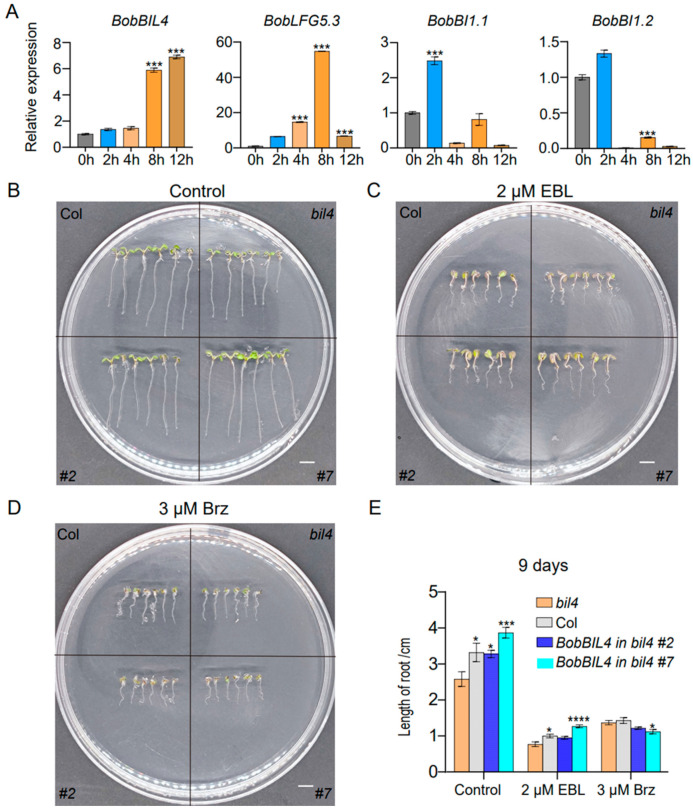
*BobBI-1* expression patterns following an exposure to an inducer or inhibitor of brassinolide stress as well as the involvement of *BobBIL4* in the induction and inhibition of brassinolide stress. (**A**) Roots were collected at 0, 2, 4, 8, and 12 h following the initiation of the 2 μM epibrassinolide (EBL) treatment. (**B**–**D**) Images of wild-type, atbil4 mutant, and *BobBIL4*-overexpressing Arabidopsis seedlings grown on normal MS medium, MS medium containing 2 μM EBL, and MS medium containing 3 μM brassinazole (Brz) for 9 days. (**E**) Root length. Error bars indicate mean ± SD (n = 15). Asterisks signify significant differences among samples (α = 0.05, Duncan’s multiple range test, * *p* < 0.05, *** *p* < 0.001, **** *p* < 0.0001). Scale bars = 1 cm (**B**–**D**).

**Figure 6 ijms-25-09562-f006:**
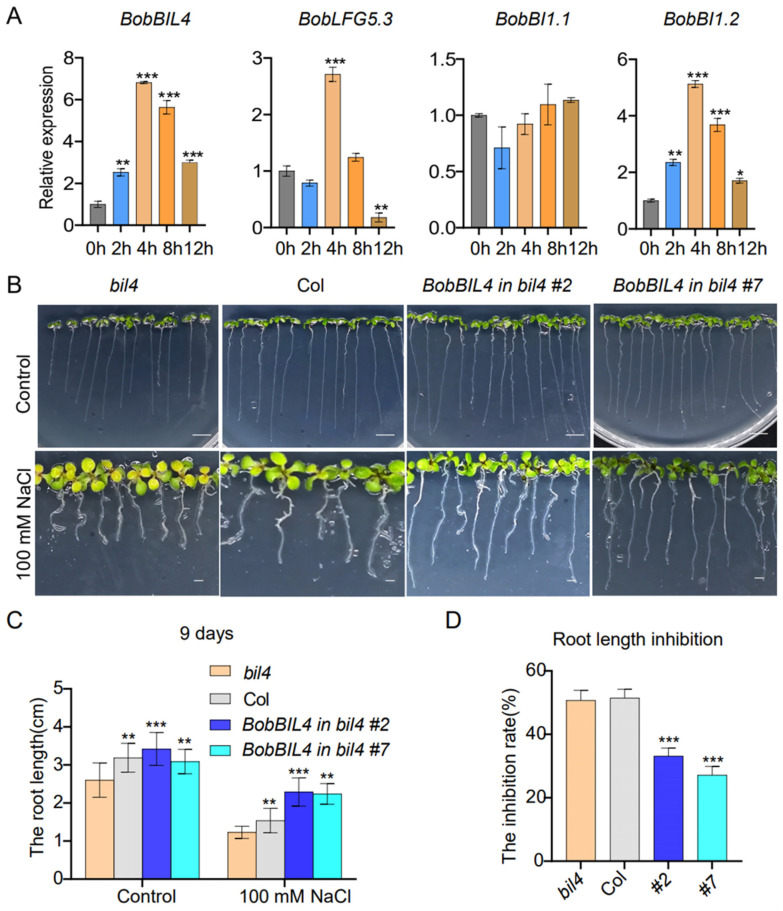
*BobBI-1* expression patterns following an exposure to NaCl stress as well as the involvement of *BobBIL4* in the response to NaCl stress. (**A**) Roots were collected at 0, 2, 4, 8, and 12 h following the initiation of the NaCl treatment. (**B**) Images of wild-type, *atbil4* mutant, and *BobBIL4*-overexpressing Arabidopsis seedlings grown on normal MS medium and MS medium containing 100 mM NaCl for 9 days. (**C**) Root length. (**D**) Root growth inhibition rate. Error bars indicate mean ± SD (n = 15). Asterisks signify significant differences among samples (α = 0.05, Duncan’s multiple range test, * *p* < 0.05, ** *p* < 0.01, *** *p* < 0.001). Scale bar = 0.7 cm in the control group; Scale bar = 0.25 cm in the 100 mM NaCl treatment group. Scale bar = 0.7 cm in the control group; Scale bar = 0.25 cm in the 100 mM NaCl treatment group.

**Table 1 ijms-25-09562-t001:** Biochemical properties of proteins encoded by *BI-1* genes.

Gene ID	Accession Number	Amino Acid (bp)	Molecular Weight (kDa)	pI	Instability Index (II)	Aliphatic Index	GRAVY	Subcellular Localization ^1^	Signal Peptide
*BobLFG1*	BolK_1g28270	277	30,941.96	6.36	28.18	128.66	0.932	GA	NO
*BobLFG2*	BolK_4g29470	248	28,079.38	9.05	26.49	125.77	0.831	C	NO
*BobLFG3*	BolK_2g39070	247	27,526.34	8.94	46.95	107.77	0.474	C	NO
*BobLFG5.1*	BolK_7g47640	239	26,980.26	7.67	24.72	130.50	0.971	CM	NO
*BobLFG5.2*	BolK_7g47660	248	27,949.24	8.27	30.42	127.38	0.877	GA	NO
*BobLFG5.3*	BolK_8g15090	256	28,111.26	5.88	26.37	128.28	0.753	GA	NO
*BobLFG4*	BolK_5g00210	256	28,111.26	5.88	26.37	128.28	0.753	GA	NO
*BobBI1.1*	BolK_2g48750	256	28,215.43	5.88	28.75	128.28	0.757	GA	NO
*BobBI1.2*	BolK_9g27570	247	27,503.40	6.89	44.91	108.58	0.500	CM, C, GA	NO

^1^ GA, Golgi apparatus; C, chloroplast; CM, cell membrane.

## Data Availability

All data are included in the article. RNA-seq raw data are available at the NCBI BioProject database under accession number PRJNA546441.

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
