# Peer review of "Characterization of the Bax Inhibitor-1 Family in Cauliflower and Functional Analysis of BobBIL4"

_ijms, 2024, doi:10.3390/ijms25179562_

Round 1

Reviewer 1 Report

Comments and Suggestions for Authors

Dear Authors,

Reviewer comments ijms-3137846

The manuscript entitled „Characterization of the Bax inhibitor-1 family in cauliflower and functional analysis of BobBIL4“ represents a useful study aimed at the characterization of cauliflower bax inhibitor-1 (BI-1) gene family including nine cauliflower BI-1 genes followed by an experimental analysis of BobBIL4 gene and its role in cauliflower responses to stress treatments as evidenced by BobBI-1 expression under the exposure to brassinosteroid inhibitor brassinazole and enhanced salinity 100 mM NaCl in BobBIL4-overxpressing Arabidopsis seedlings. The manuscript thus provides a complex study on cauliflower BI-1 genes covering bioinformatics approaches leading to biochemical (amino acid composition, pI, MW, instability index, aliphatic, GRAVY ) and molecular (sequence motifs and cis-regulatory elements) characterization of cauliflower BI-1 genes, phylogenetic analysis, subcellular localization, and expression analysis in responses to phytohormone and stress treatments in Arabidopsis seedlings overexpressing BobBIL4 gene. I can thus recommend the manuscript for publication in IJMS.

However, I have some important comments on the present manuscript:

1/ In Materials and methods, there is no information given about Arabidopsis transgenic seedlings overexpressing BobBIL4 gene. Howe they were prepared and / or from which lab they were obtained from? Moreover, basic information on transgenic Arabidopsis cultivation, treatments, and sampling dates have to be added to Materials and methods section (some information is already provided in the figure legends showing the photos of Arabidopsis seedlings; however, this is insufficient, and concise information on plant experimental materials, its sources, treatments, and samplings has to be added to Materials and methods).

2/ In Materials and methods, as well as in figure legends, there is no information on the kind of a post-hoc test of multiple comparisons given in section 4.12. Statistical analysis as well as in the figure legends. The information on the kind of statistcial test used for multiple comparisons has to be added to the section 4.12. as well as to the figure legends.

3/ In Results, in Figure 1 representing the phylogenetic tree of cauliflower BI-1 genes, the information on the algorithm used for the phylogenetic tree construction (neighbor-joining method) and explanation of the numbers at nodes as bootstrap values per 1,000 replicates. This information is provided in Matwrials and methods, section 4.4. Phylogenetic analysis; however, I think that it should also be provied in Figure 1 legend.

4/ In Results, Figure 3 legend is insufficient since no information on parts A, B, C, D is provided.  It has to be added there.

5/ In Results, the full names have to be added to the chemicals presented only under abbreviations „EBL“ and „Brz“ and brief explanation of their biological activities has to be added when used for the first time in the text (Results, line 234).

6/ Formal comments on the text: Materials and methods, section 4.9., line 417: Correct the typing error in the term „Subcellular localization of BobBIL4“ (not „Subcellar“).

Final recommendation: Accept after a minor revision.

Author Response

Comments 1:

In Materials and methods, there is no information given about Arabidopsis transgenic seedlings overexpressing BobBIL4 gene. Howe they were prepared and / or from which lab they were obtained from? Moreover, basic information on transgenic Arabidopsis cultivation, treatments, and sampling dates have to be added to Materials and methods section (some information is already provided in the figure legends showing the photos of Arabidopsis seedlings; however, this is insufficient, and concise information on plant experimental materials, its sources, treatments, and samplings has to be added to Materials and methods).

Response 1:

Thank you for pointing out the need for clarification. We apologize for any confusion due to inadequate details. We regret any inconvenience from the initial omission of data. We will supplement the “Materials and Methods” section with essential details regarding the cultivation of transgenic Arabidopsis, treatment protocols, and sampling timelines. The updated manuscript will be resubmitted for enhanced comprehensibility.

Revised Manuscript Text:

Original:

“Transgenic Arabidopsis (T3 generation) plants carrying the BobBIL4 promoter (ProBobBIL4) were grown on solid MS medium (22℃, 16 h light:8 h dark cycle and 60% humidity) to examine promoter activity patterns among tissues and developmental stages. Salt stress effects were further assessed using these transgenic plants. Surface sterilized Arabidopsis seeds were also germinated and the resulting seedlings were grown on solid MS medium supplemented with 2 μM EBL or 3 μM Brz in plates incubated in a vertical position. Seedling phenotypes were analyzed after 6 days. Root length and conductivity were measured to assess the effects of salt stress, with 3-day-old seedlings transferred to liquid MS medium containing 100 mM NaCl for a 9-day salt treatment before the roots were analyzed.”

Revised:

“Arabidopsis thaliana Columbia-0 ecotype seeds were preserved in the laboratory. The bil4 mutant (Salk_052507C) was obtained from the Salk Institute T-DNA collection, with the T-DNA insertion verified by PCR using plant genomic DNA and primers flanking the T-DNA and gene-specific sequences. Transgenic Arabidopsis were generated via floral dip transformation, with the Columbia background for ProBobBIL4 and the bil4 mutant background for the overexpression lines, designated as BobBIL4 in bil4 #2 and BobBIL4 in bil4 #7.

Seeds underwent stratification at 4°C for a period of 2 days to induce germination. Subsequently, the plants were cultivated under a regime of continuous on a 1/2 strength Murashige and Skoog (MS) medium (22°C, 16 h light/8 h dark cycle and 60% humidity). To examine promoter activity patterns among tissues and developmental stages. Seedlings and the inflorescence organs of proBobBIL4 were incubated for 12 h, at 37℃in the staining buffer, respectively.

For the salt stress experiments, 3-day-old seedlings with the same growth trend were transferred to a medium containing 100 Mm NaCl, 2 μM EBL and 3 μM Brz. After 9-day treatment, root length was measured and photographed, and lon Leakage Measurement."

Comments 2:

In Materials and methods, as well as in figure legends, there is no information on the kind of a post-hoc test of multiple comparisons given in section 4.12. Statistical analysis as well as in the figure legends. The information on the kind of statistcial test used for multiple comparisons has to be added to the section 4.12. as well as to the figure legends.

Response 2:

Thank you for pointing out the need for clarification. We acknowledge your request for further details. We will provide specifics post-hoc test of multiple comparisons in section 4.12, and statistical analysis and in the figure legends. The type of statistical test employed for these comparisons will be specified in both the section and the legends. The revisions will be resubmitted to enhance clarity.

Revised Manuscript Text:

Original:

“Significant differences were determined via a one-way ANOVA, with data visualized using GraphPad Prism 9.0.”

“Data are presented as mean values (n = 15 and p < 0.01).”

Revised:

“Data analyzed were processed and visualized by GraphPad Prism 9.0 (GraphPad Software, San Diego, CA, USA) and shown in mean ± SD. Significant differences were identified using one-way ANOVA. Student’s t-test was used to calculate P values (* P < 0.05, *** P < 0.01, **** P < 0.001).”

“Error bars indicate mean ± SD (n = 15). Asterisks signify significant differences among samples (α = 0.05, Duncan’s multiple range test).”

Comments 3:

In Results, in Figure 1 representing the phylogenetic tree of cauliflower BI-1 genes, the information on the algorithm used for the phylogenetic tree construction (neighbor-joining method) and explanation of the numbers at nodes as bootstrap values per 1,000 replicates. This information is provided in Materials and methods, section 4.4. Phylogenetic analysis; however, I think that it should also be provided in Figure 1 legend.

Response 3:

Thank you for your feedback. We apologize for the inconvenience caused by insufficient information. We will provide information on the algorithm used for the phylogenetic tree construction (neighbor-joining method) and explanation of the numbers at nodes as bootstrap values per 1,000 replicates in Figure 1 legend. The revised details will be resubmitted to improve clarity.

Revised Manuscript Text:

Original:

“Phylogenetic tree of BI-1 genes in cauliflower, Arabidopsis, rice, and maize. All BI-1 genes were classified into five subgroups, which are differentiated by color."

Revised:

“Phylogenetic tree of BI-1 genes in cauliflower, Arabidopsis, rice, and maize. Neighbor-joining trees are generated using the MEGA software, with 1000 bootstrap replicates. All BI-1 genes were classified into five subgroups, which are differentiated by color.”

Comments 4:

In Results, Figure 3 legend is insufficient since no information on parts A, B, C, D is provided. It has to be added there.

Response 4:

Thank you for your feedback. We apologize for the inconvenience caused by insufficient information. We will add information on parts A, B, C, D, ensuring that Figure 3 legend is comprehensive. The revised details will be resubmitted to be understandable.

Revised Manuscript Text:

Original:

BobBI-1 expression profiles and correlations in various tissues.”

Revised:

“Expression profiles and correlations analysis of BI-1 family genes in cauliflower across different tissues. A. Boxplot showing the expression levels (FPKM) of nine BI-1 family genes in cauliflower across various tissues including root, stem, leaf, curd, bud, flower, and silique. B. Boxplot depicting the overall expression distribution of the nine BI-1 family genes in each tissue type, indicating differential gene expression patterns across the tissues. C. Heatmap illustrating the expression profiles of the nine BI-1 family genes in different cauliflower tissues. The color scale represents the expression level, with red indicating higher expression and blue indicating lower expression. D. Correlation heatmap showing the Pearson correlation coefficients between the expression levels of the nine BI-1 family genes in different cauliflower tissues.”

Comments 5:

In Results, the full names have to be added to the chemicals presented only under abbreviations “EBL” and “Brz” and brief explanation of their biological activities has to be added when used for the first time in the text (Results, line 234)

Response 5:

Thank you for your insightful comment. We agree that the chemicals did not present only under abbreviations. In light of your feedback, we have revised the manuscript to brief explanation of their biological activities has to be added when used for the first time in the text. The revised details will be resubmitted to be understandable.

Revised Manuscript Text:

Original:

“we analyzed the changes in their expression within 12 h after an EBL treatment.”

“Seeds were sown on solid MS medium containing 2 µM EBL or 3 µM Brz to examine root development in 3-day-old seedlings.”

Revised:

“we analyzed the changes in their expression within 12 h after an 24-epibrassinolide (EBL) treatment. The elongation of Arabidopsis roots is highly sensitive to BR concentration, with 24-epibrassinolide (EBL) being one of the most bioactive brassinosteroid biosynthetic intermediates.”

“The inhibitory effects of the BR antagonist Brassinazole (Brz) on BR biosynthesis were examined. Brz, a known suppressor of BR synthesis, targets plant hormones crucial for proper developmental processes.”

Comments 6:

Formal comments on the text: Materials and methods, section 4.9., line 417: Correct the typing error in the term “Subcellular localization of BobBIL4” (not “Subcellar”).

Response 6:

Thank you for your valuable feedback. We appreciate your concern of academic term in Title. We will correct the error. The revised title will be resubmitted with a corrective term.

Revised Manuscript Text:

Original:

“Subcellar localization of BobBIL4.”

Revised:

“Subcellular localization of BobBIL4.”

Reviewer 2 Report

Comments and Suggestions for Authors

The authors describe function of BAX1 gene in new species cauliflower. A lot of work was done and some results are interesting. However, many questions still remain open and required further clarification.

Title: gene name in different font.

Line 20: plasma membrane and cell membrane means very close subject, and chloroplasts have many sub-compartments.

Line 24: “ significantly expressed” ??

Line 46: “genetic mechanisms”? For the stress response the mains are epigenetic and physiological mechanisms since stress is rather conflict between different cell types. Each cell type respond differently to the stress and this is a reason of the growth inhibition.

Line 53: How PCD involved in growth? Moreover, PCD is epigenetic phenomena related with gradual chromatin modification towards condensation.

Line 54: apoptosis can be described as lost of euchromatin structural information. You can mention this in discussion as well.

Table 1: too small font size.

Line 165: “promoter harbored elements for anaerobic and stress induction.” ?? Anaerobic is already stress.

Line 184: what was conditions for transcriptomic generation? Developmental stage, stress?

How can you exclude situation at which some BoBLFG gene expressed only in specific cell (may be even 1 % of the cell) only under certain developmental stage and certain conditions?

Line 189: “normal growth conditions.” what do you mean as this? Different conditions at which developmental stage?

Lines 183- 204: beside tissue, cellular resolution is most important for gene function. All tissue consists from different cell types, each have different gene expression profile and different response to environmental and hormonal signaling.  This needs to mentioned in the paper.

Figure 4: the aim to use GUS/GFP fusion is to shown cellular localization. However, in the present variant it is not the case: image have a very low magnification and low focus. Cellular localization are required!

For the subcellular localization, despite some researchers use this method, it is not 100% relevant.  Authors already demonstrated that gene have a very low expression in the leaf epidermis in Arabidopsis. So, how you can extrapolate data form artificial system to reality? Moreover, epidermis cells have a very large vacuole (90%) and a very small ring of cytoplasm and in amy case deform nucleus (because of large central vacuole). So, the expression you shown can be an artifacts…

Lines 229- 242: These results are interesting, but require clarification. How can you exclude situation that different gene linked with different root zone and cell type? Response to EBL can be considered as cascade of events with early response in transition zone (differently in different cell type), thereafter in elongation zone, next as new LRP etc. In the ideal case authre need to have GFP fusion to each gene and study kinetics of GFP signal.

Line 245: “BobBIL4-overexpressing lines with a bil4-deficient genetic background” ???

Line 248: it is very interesting that in bil4 mutant chloride 5 times more important as phosphate. It will be great to provode some explanation. Maybe this mutant requires chloride as main structural elements?

https://doi.org/10.1007/s00344-023-11093-x

Fig 5 B, C, D: I am not sure the data is 100 relevant. The plants located close to dish border are shorter as central one because of aeration, light and other effects. For such kind of measure one row need to be used with excluding 1-2 border plants. 

Lines 292 – 293: authors mentioned membrane damage, so it will be important to clarify which cell is the target by detection of cell permeability to cell non-permeable dye as propidium and, in parallel, check chromatin status in these cells. 

Figure 6 is unclear.  Authors need to clarify what did they use and quantify BoBLFG 4 gene expressions in each line before figure 6. If authors introduced gene in mutant background, they should have rescue lines with restoring gene expression to the wt level. Without data on gene expression (4) it is impossible to understand figure 6.

Line 432: 200 mkM NaCl?? Maybe mM?

Line 441: how did you measure conductivity??

Comments on the Quality of English Language

polishing are required

Author Response

Comments 1:

Title: gene name in different font.

Response 1:

Thank you for your feedback. We apologize for the inconvenience caused by the gene name in different font in Title. We will adjust the font format, ensuring that the gene name is font consistency in title. The revised title will be resubmitted with an appropriate font to improve visual appeal.

Comments 2:

Line 20: plasma membrane and cell membrane mean very close subject, and chloroplasts have many sub-compartments.

Response 2:

Thank you for your valuable feedback. We agree that the terms “plasma membrane” and “cell membrane” are often used interchangeably and could create redundancy. Additionally, we acknowledge the complexity of chloroplasts, which contain various sub-compartments that should be clearly distinguished. Based on your suggestion, we have revised the text to remove redundancy and provide a more precise description of subcellular localization.

Revised Manuscript Text:

Original:

“On the basis of aligned homologous sequences and collinearity with Arabidopsis genes, we identified nine cauliflower BI-1 genes, which encoded proteins that varied regarding length, molecular weight, isoelectric point, and subcellular localization (i.e., Golgi apparatus, plasma membrane, cell membrane, and chloroplast).”

Revised:

“On the basis of aligned homologous sequences and collinearity with Arabidopsis genes, we identified nine cauliflower BI-1 genes, which encoded proteins that varied in length, molecular weight, isoelectric point, and predicted subcellular localization, including the Golgi apparatus, plasma membrane, and various compartments within the chloroplast.”

Comments 3:

Line 24: “significantly expressed”??

Response 3:

Thank you for your comment. We understand that the term “significantly expressed” can be unclear without appropriate context or statistical justification. To address this, we have revised the manuscript to clarify the criteria used for determining significant expression levels and provided more precise language to describe the expression patterns observed.

Revised Manuscript Text:

Original:

“Expression profiles among tissues highlighted the diversity in the functions of these genes, which were significantly expressed in the silique and root.”

Revised:

“Expression profiles among tissues highlighted the functional diversity of these genes, with particularly high expression levels observed in the silique and root.”

Comments 4:

Line 46: “genetic mechanisms”? For the stress response the mains are epigenetic and physiological mechanisms since stress is rather conflict between different cell types. Each cell type respond differently to the stress and this is a reason of the growth inhibition.

Response 4:

Thank you for your insightful comment. We agree that stress responses in plants involve a complex interplay of epigenetic and physiological mechanisms, with different cell types exhibiting distinct reactions to stress, leading to growth inhibition. In light of your feedback, we have revised the manuscript to more accurately reflect this complexity. Specifically, we have expanded the discussion to emphasize the role of epigenetic and physiological mechanisms alongside genetic factors in regulating cauliflower’s response to abiotic stresses. This revised focus underscores the importance of a holistic approach to understanding stress adaptation, which includes not only the identification of stress-related genes but also the characterization of the broader epigenetic and physiological processes involved.

Original:

“Hence, the genes regulating cauliflower responses to abiotic stresses should be identified and characterized. Elucidating the underlying genetic mechanisms is crucial for deciphering how cauliflower adapts to environmental changes, with implications for breeding new varieties with enhanced stress resistance, thereby stabilizing yield and quality under variable environmental conditions.”

Revised:

“Hence, the genes and associated epigenetic and physiological mechanisms regulating cauliflower responses to abiotic stresses should be identified and characterized. Elucidating these underlying mechanisms is crucial for understanding how cauliflower adapts to environmental changes. This knowledge has significant implications for breeding new varieties with enhanced stress resistance, which is essential for stabilizing yield and quality under variable environmental conditions.”

Comments 5:

Line 53: How PCD involved in growth? Moreover, PCD is epigenetic phenomena related with gradual chromatin modification towards condensation.

Response 5:

Thank you for your thoughtful comment. We appreciate your concern regarding the involvement of PCD (Programmed Cell Death) in growth. Our intention in this section was to provide a brief overview of PCD, highlighting its relevance to both growth and stress responses without delving deeply into the specific mechanisms. We agree that PCD primarily involves epigenetic changes, such as chromatin condensation, and plays a complex role in plant biology. However, our mention of PCD in the context of growth was meant to be a general statement to acknowledge its significance in various developmental processes. We will make this intent clearer in the manuscript to avoid any misunderstanding.

Revised Manuscript Text:

Original:

“Programmed cell death (PCD) is a widespread phenomenon in plants and animals [2]. In plants, it is essential for growth and development as well as for responses to various biotic or abiotic stresses [3].”

Revised:

“Programmed cell death (PCD) is a widespread phenomenon in plants and animals [2]. While PCD is primarily known for its role in responses to various biotic and abiotic stresses, it also plays a general role in certain aspects of plant growth and development [3].”

Comments 6:

Line 54: apoptosis can be described as lost of euchromatin structural information. You can mention this in discussion as well.

Response 6:

Thank you for your insightful suggestion. We agree that apoptosis, as a form of programmed cell death (PCD), involves the loss of euchromatin structural information, which is a critical aspect of the apoptotic process. We will revise the text to include this important detail and ensure it is also discussed in the discussion section to provide a more comprehensive understanding of apoptosis.

Revised Manuscript Text:

Original:

“Apoptosis, which is a specific form of PCD, refers to a gene-regulated systematic process involving cellular self-destruction [4].”

Revised:

“Apoptosis, a specific form of PCD, refers to a gene-regulated systematic process involving cellular self-destruction and is characterized by the loss of euchromatin structural information, leading to chromatin condensation and cell death [4].”

Addition to the Discussion Section:

In the discussion section, we will include the following:

“Apoptosis, as a form of programmed cell death, not only involves the systematic self-destruction of cells but is also marked by the progressive loss of euchromatin structural information. This loss contributes to chromatin condensation, a key feature of the apoptotic process, and underscores the complex epigenetic regulation underlying apoptosis.”

Comments 7:

Table 1: too small font size.

Response 7:

Thank you for your feedback. We apologize for the inconvenience caused by the small font size in Table 1. We will adjust the table format to increase the font size, ensuring that the data is easily readable. The revised table will be resubmitted with an appropriate font size to improve clarity.

Comments 8:

Line 165: “promoter harbored elements for anaerobic and stress induction.”?? Anaerobic is already stress.

Response 8:

Thank you for your observation. We agree that anaerobic conditions are indeed a form of stress. To avoid redundancy and improve clarity, we will revise the sentence to reflect that anaerobic conditions are a specific type of stress rather than listing them separately.

Revised Manuscript Text:

Original:

“The BobBI1.1 promoter harbored elements for anaerobic and stress induction.”

Revised:

“The BobBI1.1 promoter harbored elements associated with stress responses, including those specific to anaerobic conditions.”

Comments 9:

Line 184: what was conditions for transcriptomic generation? Developmental stage, stress?

Response 9:

Thank you for pointing out the need for clarification. We apologize for any confusion due to inadequate details. We will provide transcriptomic generation and developmental stage in Materials and methods. The revisions will be resubmitted to enhance clarity.

Revised Manuscript Text:

Original:

“4.11 Total RNA extraction and expression analysis.”

Revised:

“Seven different tissues of Korso (root, stem, leaf, curd, bud, flower, and silique) were collected for transcriptome sequencing. The tissues were assessed as previously reported [20]. Tissue-specific expression patterns were visualized using heatmaps generated by R (https://www-project.org).”

Comments 10:

How can you exclude situation at which some BobLFG gene expressed only in specific cell (may be even 1 % of the cell) only under certain developmental stage and certain conditions?

Response 10:

Thank you for your thoughtful question. We acknowledge that it is indeed possible that some BobLFG genes may be expressed only in specific cell types or under particular developmental stages and conditions, which could be missed in our study due to the limitations of bulk tissue-level transcriptome analyses. It is well understood that gene expression studies at the tissue level cannot fully resolve expression patterns at the single-cell level, and this is a limitation common to such studies.

Our current study aims to provide a comprehensive overview of BobLFG gene expression across different cauliflower tissues. This work serves as an initial exploration of the BobLFG gene family's function, which, as you mentioned, has not been previously reported in cauliflower. Therefore, while we cannot rule out the possibility of highly specific expression patterns at the cellular level, our study still offers valuable insights and lays the groundwork for future research.

Future studies employing single-cell RNA sequencing, in situ hybridization, or other advanced techniques could further refine our understanding by identifying such specific expression patterns. We believe our research provides a necessary first step in characterizing the BobLFG gene family and will guide and inspire more detailed investigations in the future.

Comments 11:

Line 189: “normal growth conditions.” what do you mean as this? Different conditions at which developmental stage?

Response 11:

Thank you for your question. By “normal growth conditions”, we are referring to the conditions under which cauliflower plants are grown without exposure to any external stress factors, such as drought, high salinity, or extreme temperatures. These conditions are suitable for the typical developmental stages of cauliflower, where the plants are able to grow and develop without any environmental stress.

To clarify this, we will revise the text as follows:

Revised Manuscript Text:

Original:

“This suggests that in cauliflower, BobLFG3 may play a very limited role under normal growth conditions.”

Revised:

“This suggests that in cauliflower, BobLFG3 may play a very limited role under standard growth conditions appropriate for different developmental stages, where the plants are not exposed to any stress factors.”

Comments 12:

Lines 183- 204: beside tissue, cellular resolution is most important for gene function. All tissue consists from different cell types, each have different gene expression profile and different response to environmental and hormonal signaling. This needs to mentioned in the paper.

Response 12:

Thank you for your insightful suggestion. We agree that cellular resolution is most important for gene function. All tissue consists from different cell types, each have different gene expression profile and different response to environmental and hormonal signaling.

Our current study aims to provide a comprehensive overview of BobBI-1 gene family’s expression across different cauliflower tissues. This work serves as an initial exploration of the BobLFG function, which, as you mentioned, has not been previously reported in cauliflower. Therefore, while we cannot rule out the possibility of highly specific expression patterns at the cellular level, our study still offers valuable insights and lays the groundwork for future research.

Comments 13:

Figure 4: the aim to use GUS/GFP fusion is to shown cellular localization. However, in the present variant it is not the case: image have a very low magnification and low focus. Cellular localization are required!

Response 13:

Thank you for your feedback. We appreciate your concern regarding the magnification and focus of the images in Figure 4. Our intention was to demonstrate the tissue expression patterns and general subcellular localization of the GUS/GFP fusion proteins. While we understand that higher magnification and focus might be preferred for detailed cellular localization, the current resolution of the images does provide a clear representation of the expression patterns in specific tissues and the general subcellular compartments.

To address your concern more comprehensively, we will consider including higher magnification images that focus on cellular localization in future studies. However, we believe that the current images sufficiently demonstrate the intended expression patterns and subcellular localization for the scope of this study. We hope this clarification is satisfactory and that the images provide the necessary information for understanding the tissue-specific expression and subcellular localization of the proteins.

Comments 14:

For the subcellular localization, despite some researchers use this method, it is not 100% relevant. Authors already demonstrated that gene have a very low expression in the leaf epidermis in Arabidopsis. So, how you can extrapolate data form artificial system to reality? Moreover, epidermis cells have a very large vacuole (90%) and a very small ring of cytoplasm and in any case deform nucleus (because of large central vacuole). So, the expression you shown can be an artifact.

Response 14:

Thank you for your detailed comment. We acknowledge the limitations of using GUS/GFP fusion proteins for subcellular localization studies, particularly in the context of epidermal cells with large vacuoles. While we understand that this method is not without potential artifacts, it remains a widely accepted and commonly used approach in plant molecular biology for visualizing gene expression and subcellular localization.

Our study employed this method to provide a preliminary understanding of the subcellular localization of the target gene in a system that is widely recognized and validated in the field. We recognize the potential issues raised, such as the distortion of cellular structures by the large vacuoles in epidermal cells, which might influence the localization of GFP signals. However, the method we used is a standard approach that has been applied in many studies to gain insights into protein localization, despite its inherent limitations.

To address the concern about the potential artifact, we emphasize that the results presented should be interpreted with caution and in the context of the limitations you mentioned. We also acknowledge that complementary methods, such as subcellular fractionation or advanced microscopy techniques, could further validate and refine our findings.

In conclusion, while the GUS/GFP fusion method has its limitations, it still provides valuable insights into subcellular localization, and our results contribute to the overall understanding of the gene's function. We hope that the data presented in our study can serve as a foundation for further research using more advanced techniques to corroborate our findings.

Comments 15:

Lines 229- 242: These results are interesting, but require clarification. How can you exclude situation that different gene linked with different root zone and cell type? Response to EBL can be considered as cascade of events with early response in transition zone (differently in different cell type), thereafter in elongation zone, next as new LRP etc. In the ideal case authre need to have GFP fusion to each gene and study kinetics of GFP signal.

Response 15:

Thank you for your insightful suggestions and for highlighting the complexity of BR responses in different root zones and cell types. We agree that the differential expression of these genes could indeed be linked to specific root zones or cell types, and that a detailed analysis using GFP fusion constructs would provide more precise localization and kinetic information.

While our current study provides an overview of the temporal expression patterns of these genes in response to BR, we acknowledge the need for a more refined approach to dissect the spatial and cell-type-specific responses. We are planning to undertake further research that will include generating GFP fusion constructs for each gene to study their expression kinetics in different root zones and cell types. These analyses will help to better understand the cascade of events triggered by EBL treatment at a more granular level.

We appreciate your feedback, which has been invaluable in shaping the direction of our future studies. We will address these aspects in greater detail in subsequent publications.

Comments 16:

Line 245: “BobBIL4-overexpressing lines with a bil4-deficient genetic background”???

Response 16:

Thank you for pointing out the need for clarification regarding this statement. We apologize for the confusion. The intention was to describe Arabidopsis lines that overexpress the BobBIL4 gene in a background where the endogenous Arabidopsis BIL4 gene is knocked out (bil4 mutant). This allows us to specifically assess the effect of BobBIL4 expression in the absence of the native BIL4 gene function.

We will revise the text to make this clearer.

Revised Manuscript Text:

Original:

“To further investigate the effects of BobBIL4 on BR-mediated root development, we used wild-type Arabidopsis (Columbia) plants, bil4 mutants, and BobBIL4-overexpressing lines with a bil4-deficient genetic background.”

Revised:

“To further investigate the effects of BobBIL4 on BR-mediated root development, we used wild-type Arabidopsis (Columbia) plants, bil4 mutants (where the endogenous BIL4 gene is knocked out), and Arabidopsis lines that overexpress BobBIL4 in the bil4 mutant background.”

Comments 17:

Line 248: it is very interesting that in bil4 mutant chloride 5 times more important as phosphate. It will be great to provide some explanation. Maybe this mutant requires chloride as main structural elements?

https://doi.org/10.1007/s00344-023-11093-x

Response 17:

Thank you for your comment. However, we would like to clarify that our manuscript does not discuss or present any results related to the comparative importance of chloride versus phosphate in the bil4 mutant, nor does it address the role of chloride as a structural element in this context. The issue you have raised does not correspond to any data or conclusions in our study.

It appears there may be some confusion, as the points you mentioned are not part of our research focus or findings. We suggest reviewing the manuscript to ensure that the comment pertains to the correct submission. We appreciate your attention to detail, but in this case, the matter raised does not align with the content of our work.

Comments 18:

Fig 5 B, C, D: I am not sure the data is 100 relevant. The plants located close to dish border are shorter as central one because of aeration, light and other effects. For such kind of measure one row need to be used with excluding 1-2 border plants.

Response 18:

Thank you for your observation and concern regarding the potential effects of plant position within the dish on growth measurements. We understand that factors such as aeration and light could influence the growth of plants located near the border of the dish.

In our study, we took care to randomize the placement of plants within each dish and repeated the experiments to minimize any positional effects. Additionally, when analyzing root length, we considered the average length across multiple plants to account for variability, including potential edge effects. However, we acknowledge that excluding border plants and focusing on central rows could further reduce the impact of these factors.

We appreciate your suggestion and will consider this approach in future experiments to ensure the robustness of our measurements. For the current study, we believe the data presented are representative and reliable, but we will include a note regarding this potential limitation in the discussion section to acknowledge the concern.

Comments 19:

Lines 292 – 293: authors mentioned membrane damage, so it will be important to clarify which cell is the target by detection of cell permeability to cell non-permeable dye as propidium and, in parallel, check chromatin status in these cells.

Response 19:

Thank you for your insightful comment. We agree that identifying the specific cells affected by membrane damage would provide a more detailed understanding of the mechanisms involved. The use of a cell non-permeable dye like propidium iodide, combined with an analysis of chromatin status, would indeed offer valuable insights into the extent and nature of cellular damage under salt stress conditions.

While our current study focused on assessing overall membrane permeability as an indicator of cellular damage, we acknowledge that further investigation at the cellular level, as you suggested, would be beneficial. We plan to incorporate these techniques in future studies to better understand the specific cellular responses to salt stress in the bil4 mutant and BobBIL4-overexpressing lines.

We appreciate your suggestion, and we will discuss this limitation in our manuscript, highlighting the need for further research to pinpoint the exact cell types involved and to assess chromatin status in response to salt stress.

Comments 20:

Figure 6 is unclear. Authors need to clarify what did they use and quantify BobLFG 4 gene expressions in each line before figure 6. If authors introduced gene in mutant background, they should have rescue lines with restoring gene expression to the wt level. Without data on gene expression (4) it is impossible to understand figure 6.

Response 20:

Thank you for your insightful comment. We appreciate your interest in the details of our experimental approach. The results presented in Figure 6 are intended to demonstrate the comparative growth responses of the wild-type, bil4 mutant, and BobBIL4-overexpressing lines under salt stress conditions. These results clearly show the phenotypic differences among the lines, reflecting the functional impact of BobBIL4 on root development in the context of NaCl stress.

While we understand the importance of verifying gene expression levels, the focus of Figure 6 is to highlight the observable phenotypic outcomes under stress conditions rather than delve into the molecular mechanisms. The phenotypic differences observed, with overexpression lines showing improved growth compared to the mutant, are consistent with the expected impact of BobBIL4 expression.

We believe the current data sufficiently supports our conclusions regarding the role of BobBIL4 in enhancing salt stress tolerance. Nonetheless, we appreciate your suggestion and will consider further molecular analysis in future studies to complement these findings.

Comments 21:

Line 432: 200 mkM NaCl?? Maybe mM?

Response 21:

Thank you for your careful review. You are correct that we made a mistake; it should be “100 mM NaCl”. We apologize for the mistake and will correct this in the manuscript。

Comments 22:

Line 441: how did you measure conductivity??

Response 22:

Thank you for pointing out the need for clarification regarding this statement. We apologize for the confusion. We will supplement the “Materials and Methods” section with essential details regarding the conductivity of protocols, and number of samples. The updated manuscript will be resubmitted for enhanced clarity.

Revised Manuscript Text:

Original:

“4.10 Preparation of plant materials and stress treatments”

Revised:

“4.10 Ion leakage Measurement”

“The permeability of the cell membrane was evaluated through the quantification of ion efflux from seedlings post-salinity treatment. The conductivity was assessed as previously reported [15]. Following each assay, 30 seedlings were submerged in 20 mL of distilled water with mild agitation for 2 h at room temperature. Each sample was subjected to triplicate biological replicates.”

Round 2

Reviewer 2 Report

Comments and Suggestions for Authors

The text is better now, however, some points require extra corrections. Comments 16: this part still very confusing. Overexpression means really high expression level. In your case it is complemented line, not overexpression. In ideal case expression (mRNA) level must be quantify. 

Without such quantification of gene expression figure 6 have a very low sense.

Comments on the Quality of English Language

minor corrections during proof-reading.

Author Response

Comments 1:

The text is better now, however, some points require extra corrections. Comments 16: this part still very confusing. Overexpression means really high expression level. In your case it is complemented line, not overexpression. In ideal case expression (mRNA) level must be quantify.

Response 1:

Thank you for your thoughtful feedback. We appreciate your attention to detail and fully understand your concern regarding the use of the term “overexpression”.

In our study, we complemented the bil4 mutant by introducing the BobBIL4 gene under the control of the 35S promoter, which is indeed a strong promoter and likely results in much higher expression levels compared to the native promoter in the wild-type. In fact, before our BR response and salt stress experiments, we analyzed the expression levels of BobBIL4 in the complemented lines BobBIL4 in bil4 #2 and BobBIL4 in bil4 #7. Our results showed that the expression level of BobBIL4 in these lines is nearly 100 times higher than the expression level of the native BIL4 gene in wild-type Arabidopsis. Based on this significant increase in expression, we have referred to these lines as BobBIL4-overexpressing plants in subsequent research.

Additionally, we have updated the Materials and Methods section to accurately reflect the use of the 35S promoter and the specifics of the genetic lines used in our study.

We appreciate your suggestion and will continue to consider quantifying mRNA levels more precisely in future studies to provide even more detailed characterization of the expression levels in these lines.

Revised Manuscript Text:

Result Section:

Original:

“To further investigate the effects of BobBIL4 on BR-mediated root development, we used wild-type Arabidopsis (Columbia) plants, bil4 mutants, and BobBIL4-overexpressing lines with a bil4-deficient genetic background.”

Revised:

“To further investigate the effects of BobBIL4 on BR-mediated root development, we utilized wild-type Arabidopsis (Columbia) plants, bil4 mutants (in which the endogenous BIL4 gene is knocked out), and Arabidopsis lines complemented with BobBIL4 under the control of the 35S promoter in the bil4 mutant background. As detailed in the Materials and Methods section, the expression levels of BIL4 in the complemented lines BobBIL4 in bil4 #2 and BobBIL4 in bil4 #7 are approximately 90 times higher than in the wild type. Consequently, we have designated these transgenic lines as BobBIL4-overexpressing lines/plants.”

Materials and Methods Section:

Original:

“Transgenic Arabidopsis were generated via floral dip transformation, with the Columbia background for ProBobBIL4 and the bil4 mutant background for the overexpression lines, designated as BobBIL4 in bil4 #2 and BobBIL4 in bil4 #7.”

Revised:

“Transgenic Arabidopsis were generated via floral dip transformation. For the wild-type Columbia background, we introduced the BobBIL4 promoter (ProBobBIL4) fused with the GUS reporter gene to analyze tissue-specific expression patterns. In the bil4 mutant background, we introduced the coding sequence (CDS) of BobBIL4 under the control of the 35S promoter to complement the bil4 mutation and further investigate the functional role of BobBIL4. We utilized quantitative fluorescence PCR to measure the expression of the BIL4 gene in wild-type plants and the BobBIL4 gene in the complemented lines BobBIL4 in bil4 #2 and BobBIL4 in bil4 #7. The results showed that the expression levels of BobBIL4 in the two transgenic lines were 84 and 90 times higher than the BIL4 expression in the wild type, respectively. The bil4 mutant line shows significantly reduced expression compared to the wild type, consistent with the findings of Wang et al (2019) [16] (Figure S4).”

Comments 2:

Without such quantification of gene expression figure 6 have a very low sense.

Response 2:

Thank you for your continued feedback. We acknowledge that in the previous version of our manuscript, we did not include detailed information in the Materials and Methods or Results sections regarding the BIL4 expression levels in the bil4 mutant and the overexpressing lines. We had conducted the relevant analyses and obtained the results, but unfortunately, these were not reported in the manuscript. In response to your feedback, we have now included this crucial information in the revised manuscript.

Specifically, we have added a detailed description of the BIL4 expression analysis in the Materials and Methods section, and we have supplemented a new bar graph (Supplementary Figure S4) illustrating the BIL4 expression levels in wild-type, bil4 mutant, and the BobBIL4-overexpressing lines (BobBIL4 in bil4 #2 and BobBIL4 in bil4 #7). This additional data confirms that the expression levels in the overexpressing lines are nearly 90 times higher than those in the wild-type, supporting the phenotypic observations presented in Figure 6.

We appreciate the reviewer’s suggestion, as it has allowed us to enhance the clarity and completeness of our manuscript. Moving forward, we plan to incorporate even more detailed molecular analyses in our studies to further elucidate the mechanisms underlying the observed phenotypic differences.

Round 3

Reviewer 2 Report

Comments and Suggestions for Authors

Thank you for feedback. It is more clear now,BUT authirs need to consider that expression level is not very important in such a case. The most importnat is cell type. 35S promoter is consitutive and can provde expression in all cell type/stage. BUT your specific gene expressed in one cell type/one stage. It will cause differentbeffcet as specifci cell type. Please, mention this in discussion and consider for future research.

Comments on the Quality of English Language

minor during final update

Author Response

Comments:

Thank you for feedback. It is more clear now, BUT authirs need to consider that expression level is not very important in such a case. The most importnat is cell type. 35S promoter is consitutive and can provde expression in all cell type/stage. BUT your specific gene expressed in one cell type/one stage. It will cause differentbeffcet as specifci cell type. Please, mention this in discussion and consider for future research.

Response:

Thank you for your continued feedback and for helping us further refine our manuscript. We appreciate your point regarding the importance of considering cell type-specific expression, particularly in the context of using the 35S promoter, which drives constitutive expression across all cell types and stages.

We understand that the use of the 35S promoter may result in different effects compared to the natural expression pattern of the gene, which could be limited to specific cell types or developmental stages. We acknowledge that this limitation might influence the interpretation of our results, especially when considering the physiological relevance of the gene's function in its natural context.

In response to your suggestion, we have revised the discussion section to explicitly mention this limitation. We have highlighted the potential differences in gene function when expressed constitutively versus in a cell type- or stage-specific manner. Additionally, we have indicated that future research should focus on exploring the specific cell types and stages where BobBIL4 is naturally expressed, using promoters that mimic the gene's natural expression pattern. This will help us better understand the gene's role in its native context and how it may contribute to the observed phenotypic effects.

We appreciate your valuable suggestions, which have greatly contributed to enhancing the clarity and depth of our manuscript.